# Two-Step Geometry Design Method, Numerical Simulations and Experimental Studies of Bioresorbable Stents

**DOI:** 10.3390/ma15072385

**Published:** 2022-03-24

**Authors:** Natalia Molęda, Grzegorz Kokot, Wacław Kuś, Michał Sobota, Jakub Włodarczyk, Mateusz Stojko

**Affiliations:** 1Faculty of Mechanical Engineering, Silesian University of Technology, 18A Konarskiego St., 44-100 Gliwice, Poland; natalia.moleda@polsl.pl (N.M.); waclaw.kus@polsl.pl (W.K.); 2Centre of Polymer and Carbon Materials, Polish Academy of Sciences, 34 Curie-Sklodowska St., 41-819 Zabrze, Poland; msobota@cmpw-pan.edu.pl (M.S.); jwlodarczyk@cmpw-pan.edu.pl (J.W.); mstojko@cmpw-pan.edu.pl (M.S.)

**Keywords:** bioresorbable stents, two-step modeling, numerical simulation, digital image correlation

## Abstract

The stent-implantation process during angioplasty procedures usually involves clamping the stent onto a catheter to a size that allows delivery to the place inside the artery. Finding the right geometrical form of the stent to ensure good functionality in the open form and to enable the clamping process is one of the key elements in the stent-design process. In the first part of the work, an original two-step procedure for stent-geometry design was proposed. This was due to the necessary selection of a geometry that would provide adequate support to the blood-vessel wall without causing damage to the vessel. Numerical simulations of the crimping and deployment processes were performed to verify the method. At the end of this stage, the optimal stent was selected for further testing. In addition, numerical simulations of selected experimental tests (catheter-crimping process, compression process) were used to verify the obtained geometrical forms. The results of experimental tests on stents produced by the microinjection method are presented. The digital image correlation (DIC) method was used to compare the results of numerical simulation and experimental tests. The two-step modeling approach was found to help select the appropriate geometry of the expanded stent, which is an extremely important step in the design of the crimping process. In the part of the paper where the results obtained by numerical simulation were compared with those gained by experiment and using the DIC method, a good compatibility of the displacement results can be observed. For both longitudinal and transverse (pinch) stent compression, the results practically coincide. The paper presents also the application of the DIC method which significantly expands the research possibilities, allowing for a detailed inspection of the deformation state and, above all, verification of local dangerous areas. This approach significantly increases the possibility of assessing the quality of the stents.

## 1. Introduction

According to data available on the World Health Organization website, the highest mortality rate (around 27%) is related to cardiovascular disease. These are two diseases at the top of the ranking: stroke (second place) and ischemic heart disease (first place) [1]. Ischemic heart disease can be caused by several causes, among which we distinguish coronary-artery stenosis, resulting from diseases that affect the entire body, such as syphilis or lupus erythematosus; or coronary-artery embolism, generally caused by bacterial infection or thrombi, or storage of abnormal metabolic products in the walls of the coronary arteries. However, the main cause, which accounts for almost 100% of all causes of coronary heart disease is artery atherosclerosis, during which the vessel narrows. This is caused by the formation of cholesterol and inflammatory cells in the blood vessels. In the final stage of the disease, this leads to the closure of the vessel [2]. It can be atherosclerosis of both the coronary and peripheral vessels. Although coronary atherosclerosis usually leads to ischemic heart disease, carotid-artery blockage caused by atherosclerosis may lead to stroke [3,4]. Furthermore, atherosclerosis in one vessel has been shown to cause atherosclerosis in another vessel [5]. According to a study, 22% of cases of atherosclerosis are peripheral-artery disease (PAD) [6], according to other studies, PAD occurs in almost half of coronary-artery diseases and in more than a third of cardiovascular diseases [7,8]. In conclusion, peripheral-artery disease is strongly correlated with other vascular diseases [9,10]. Furthermore, despite stable coronary-artery disease, PAD causes high mortality rates [11].

To prevent this, a procedure called angioplasty is performed that expands the blood vessels. The first successful treatment of this kind was carried out in the late 1970s. Unfortunately, the treatments initially performed were associated with a frequent (more than 30%) restenosis phenomenon, which consists of narrowing the blood vessel again [12,13,14].

To reduce the appearance of restenosis, vascular prostheses called stents were introduced. Their task was to support the walls and widen the lumen of the blood vessel. Initially, stents were made of metal with shape memory; it was a nickel alloy with titanium invented in the Naval Ordinance Laboratory and named NiTINOL [15,16]. Stents of this type are called bare-metal stents (BMS).

The introduction of this invention into the artery has reduced the number of acute occlusions but has not yet addressed the problem of restenosis (caused by excessive neointimal growth) and thrombosis [17]. In addition, there are problems with tissue inflammation, loosening, and the risk of fracture [18]. Due to the fact that stent-implementation surgery is associated with complications, pharmacotherapy is necessary. The use of an antiplatelet drug and a drug that inhibits platelet activity has contributed to lowering the risk of thrombosis, reducing restenosis, revascularization, and neointimal-tissue growth [19,20]. These stents are generally made of a nonresorbable material coated with a gradually released drug. They are called drug-eluting stents (DES) and were invented in 2000 [21,22].

The advantage of this type of stent is high conformability and good flexibility [23]. However, despite the new generation of stents, complications persist after the procedure. These include impaired vasomotor function of the artery, inflammatory reactions, or an increased risk of very late thrombosis. In addition, BMS and DES remain in the body throughout life. Therefore, a different technology began to emerge to eliminate these complications, resulting in the creation of bioresorbable stents (BRS). The idea of this type of stents is based on their bioresorption a few or several months after implantation [24]. There are several types of BRS. We include in them, i.a., bioresorbable stent with platinum chromium and bioabsorbable magnesium stent with the addition of rare earth metals—the first metallic stent that underwent the structure but unfortunately caused high restenosis spacers and stents made on the basis of poly-l-lactic (PLLA) [25,26].

This is possible due to the construction of these stents. The publication of research on applications in pig coronary arteries of a metal stent coated with PLLA contributed to the use of this material for stents [27]. The ideal cycle of angioplasty with a BRS stent implies the implantation of a stent that will withstand all stresses in the body (mainly radial force and flow in the blood vessel) and then dissolve after 6 months, leaving the intact artery wall and lack of diffuse conditions. This is caused by gradual corrosion caused by body fluids [28].

Poly-L-lactic acid has an ordered structure consisting of segments bound by amorphous-polymer chains. The distribution of bioresorbable stents is possible because of the phenomenon of hydrolysis. As a result of this process, the molecular weight of the material is reduced and finally disintegrated into water, carbon dioxide, and carbon [29,30].

It is their complete biodegradation that is the biggest advantage of BRS. In addition, they are characterized by regulation of vascular tone, late luminal gain, and late expansive remodeling. Due to their flexibility, they also allow normal operation of the artery—its expansion and contraction [31]. Otherwise, BRS characteristics associated with the restoration of functional endothelial cells and the absence of any foreign material can result in a lack of the need for long-term dual-antiplatelet therapy (DAPT) and reduce the risk of stent thrombosis [23,32]. However, it also has disadvantages compared to DES, such as the tendency to cause elastic recoil or lower radial support, resulting in the need to design stents with a thicker wall and higher molecular weight. In addition, using too much material to produce a stent can cause tissue deposition or a toxic reaction [33]. Unfortunately, due to the increase in this dimension of the stent, there may be a disturbance of blood flow in the vessels, as well as restenosis and thrombosis. Several strategies have been proposed to avoid having to increase the thickness of the stent and improve radial support. The first of these is the use of polymer processing: blow molding and heat treatment. The other method is to change the composition of the polymer to obtain a semicrystalline PLLA of high molecular weight. The third involves looking for other polymers that can be used on BRS stents [34].

When designing a stent, it is important to remember how the stent is placed and what adverse events may occur during this process. One of them may be dogboning, seeking more expansion at the ends of the stent than in the middle, as well as foreshortening, which causes stress accumulation [35].

This paper focuses on designing the geometry of a bioresorbable stent made of PLLA so that the thickness of the stent wall is as small as possible; however, the remaining geometric dimensions ensure adequate resistance to the crimping loads that occur during the crimping process of the stent. The load created by the closing cylindrical rigid plate (a crimping tool) cannot cause loss of stator stability due to insufficient stiffness of the object. The paper also addresses the issue of comparing the numerical simulation results with the experimental results. The stent designed by the method proposed in the first part of the article was manufactured using injection-moulding technology [36] and experimental tests were performed using a strength-testing machine combined with digital image-correlation equipment. In addition, analysis that matches the experimental studies was carried out using finite-element-method (FEM) software. This study allowed us to determine whether the results obtained in the numerical simulation actually correspond to the real stent response to the given load. Section 3 describes the design process of the stent geometry. The crimping and deployment of the stents are discussed. Section 4 is devoted to the description of the experimental setup and the boundary conditions and parameters of the numerical methods. The results of two experiments are presented in Section 5 and compared with numerical results obtained using FEM.

## 2. Materials

Poly-L-lactide (PLLA) with a molecular weight of 250 kD using the size-exclusion-chromatography (SEC) method were obtained at the Center for Polymer and Carbon Materials. Monomer: L-lactide (LA) from Huizhou Foryou Medical Devices Co., Ltd., Huizhou, China, was purified by recrystallization from anhydrous ethyl acetate and then dried in a vacuum oven at room temperature to a constant weight. The initiator in the process was zinc (II) acetylacetonate monohydrate, which was from Avantor, Poland. LA polymerization was carried out in bulk at 140 °C. [Zn[(acac)_2_∙H_2_O] with an initiator/monomer ratio (I/M) of 1/600 was used as an initiator for this reaction.

## 3. Methods

The typical design procedure is to create the stent-geometry model of the shape of the stent corresponding to its open form. However, one of the important stages of the angioplasty process is the stage of the stent insert in the affected artery. At this stage, the stent is crimped in closed form to deliver it to the appropriate position in the artery. We observe that in many cases the typical designing process leads to shapes which are difficult or impossible to crimp. The proposed designing procedure starts with the shape of the closed form. Based on the design experience obtained during work on the stent shape, where the shape of the stents is generally designed as their open forms, the new two-step modeling procedure is proposed to design the stent. The original part of this method is to reverse the typical design process for this kind of object.

The compact closed form undergoes the numerical simulation of the expansion process to the final open form. On the basis of our experience, this reversed procedure is much more efficient in the stent-design process. The function snap is shown in Figure 1.

Starting with an initial design of the closed form of a stent, the expending process to the open form is numerically simulated. This step shows the ability of the design to open and evaluate the displacement and stress state. The open form gives information about changes that should be introduced to the initial design. Changes in the initial design lead to the finding of the shape, which also has an optimal shape in open form. This loop can also be performed using optimization procedures.

Figure 2 shows the numerical stages of the deployment process (on the left) and the crimping process (on the right). As we can see in the picture, in the case of the deployment process, the geometry—which is marked in green in the drawing—expanded, as shown in the red arrows. This process consists of multiplying the diameter of the catheter, and due to the contact between the catheter and the stent, the second one also increases its diameter (both internal and external). In the case of the crimping process, the geometry is closed by reducing the diameter of the rigid plate, and due to contact between the rigid plate and the stent, the diameter of the stent is reduced.

### 3.1. Geometric Model

In order to demonstrate the proposed design process, three initial geometries of contracted stents were designed and subjected to a numerical deployment process; then, on the basis of the expanded form of stents, the optimal form is selected, or some geometrical changes are made. The initial geometry was chosen based on preliminary research in the field. Some information on stent design was provided by experimental studies on obtaining preset shapes produced by microinjection. The selection criteria were the ability to generate an appropriate closed- and open-form stent with the appropriate displacement and stress state. The variable in the geometric models was the length of the stent crown, which was manipulated to achieve satisfactory geometry and strength properties of the stent after deployment. Other values, such as: stent length and thickness, radius value of the rounded ring, and the thickness of the link, remained the same in all cases. In Figure 3 three proposed stent geometries are presented, which in the next step were subjected to a deployment process and in which the geometry of the expanded stent was observed on the basis of the input forms of the contracted stent.

The sketch of half of the selected stent with marked characteristic values is presented in Figure 4. The thickness of the stent was 0.2 mm. This parameter was determined by experimental studies of stents obtained by microinjection using polymeric materials, as well as by simulation of the manufacturing process and the possibility of stent placement on catheter.

### 3.2. Numerical Simulation

The initial geometry models were subjected to a numerical simulation of the expansion process to their final position in the arterial space. The FEM as a well-known and proven method in the modeling of mechanical systems was used during computations [37]. As numerical simulation software, the MSC Marc Computer-Aided Engineering (CAE) system was used. The models were discretized with eight-noded hexahedral finite solid elements with quadratic-shape functions. The average mesh size for each model was 0.05 mm. The division of three geometrical models into finite elements is shown in Figure 5. It was assumed that the material is linear-elastic, and the material properties correspond to typical PLLA (Young modulus 40 MPa and Poisson ratio 0.33) [38,39].

The numerical model of the deployment process is presented in Figure 6. The FEM mesh representing the stent has a contact-boundary condition with the inner surface, which is the catheter. This surface has a diameter equal to the outer diameter of the catheter. As mentioned at the beginning of the subsection, during numerical simulation, it gradually changes to destination diameter, which depends on the open form of the stent geometry, expanding the contracted stent on occasion. The stent also has a contact-boundary condition with an external surface modeled as a rigid surface with a diameter close to the inner diameter of the vessel. As a result of properly defined boundary conditions, the extended stent is adapted to two cylinders. Their diameters of both open rigid plates differ by the two wall thicknesses of the stent, so that the stent is adequately expanded but also not deformed as a result of too much pressure on the surface.

Based on the numerical model, a deployment-process simulation was performed, and the results in the form of the maps of equivalent stresses and displacements were obtained. The maps are shown in Figure 7 and Figure 8. The unit of stress is MPa and displacement—mm.

Based on the shape of the stent after the process, it was assessed whether the geometry created previously is unique to creating the geometry of an open stent. Too large a deformation of the model or an extension of the stent crown precluded the selection of a given numerical model as the basis for small-change introduction creating a geometric model of an open stent. One of the designed stents met the requirements and it was possible to generate a modified geometrical model of the open form in the CAD system (Autodesk Inventor was used). Figure 9 presents created geometric and numerical models. In the case of the numerical model, an identical division into finite element was used. The material model is also the same, with the characteristic PLLA properties.

As with the deployment-process simulation for the numerical-simulation crimping-process model, we also use two rigid surfaces (Figure 10). In this case, the course of the numerical simulation is slightly different. The simulation consists in reducing the diameter of the outer plane almost by half, and due to the contact between it and the FEM model, the stent is also contracted. The stent against has contact with the external surface of the inner cylinder. This allows the required form of the stent to be achieved after the crimping process.

As a result, we received maps of equivalent stresses and displacements presented in Figure 11. The unit of stress is MPa and displacement—mm.

The largest deformation of a stent occurs at its ends and is 1.82 mm (Figure 11). The highest stress values are found in the place where the radius of the rounding of the stent crown is changing and is equal to 11 MPa. This value is more than three times lower than the Young module value for PLLA, which means that it will not yield to plasticization of the stent.

## 4. Numerical Simulations and Experimental Test

To compare the results between numerical simulation and experimental tests, a stent-test sample based on the proposed procedure was designed and produced using the microinjection-moulding machine (Micropowder 15, Wittman-Battenfeld, Grodzisk Mazowiecki, Poland) equipped with a highly precise mold. First, a numerical simulation of the simple compression test was performed. The set of tests (both for numerical simulations and experimental tests) was chosen as one of the cases of deformation during the implementation process. This type of test is popular and proposed, for example, by some industry manufacturers (Mechanical Properties of the Gore Tigris Vascular Stent; W. L. Gore & Associates, Inc., Warsaw, Poland [40]). One of these tests is a pinch because when a stent is placed in a blood vessel, the guide wire (on which the stent is closed) bends as it passes through veins or arteries to reach the target site. These two tests (longitudinal and transverse compression) are close to the behavior of the stent during the stenting procedure. The second reason was the possibility of comparing the numerical simulations with the experimental testing using the DIC method. The created geometry of the test sample with its discrete form (eight-noded hexahedral finite solid elements) is presented in Figure 12.

On the model prepared in this way, numerical simulations of stent compression were carried out. It was longitudinal compression and transverse compression (pinch). For both simulations, rigid plates were used to compress the stent in both radial and longitudinal directions. The numerical model has a defined contact between a rigid plate, which acts on the stent to induce compression in the corresponding axis, and an external surface, which acts as a base. The appropriate boundary conditions were also defined. In the simulations, the material model is based on mechanical properties which corresponds to the characteristic properties of PLLA. Figure 13 shows the stent with the plates acting as compression tools.

The same CAD-designed geometry, on which the numerical simulations were performed, was produced by microinjection moulding (Figure 14).

Once a physical model of the stent was obtained, longitudinal and transverse compression (pinch) tests were performed. Experimental tests were performed using an MTS test machine combined with Dantec Dynamic Digital Image Correlation (Dantec Dynamic Q400 system with Istra4D software). The DIC system is a noncontact optical-displacement and strain-measurement tool, which in the case of the small specimen has many advantages. Strain measurement consists of image recording of the specimen with a special speckle pattern during the test. The set of collected images are correlated using a special software in order to obtain displacement and strain color maps.

The deformation of the stent due to the applied force was observed using a system of two cameras. Camera-recorded images of the compression of the stent on the corresponding axis were then analyzed in Dantec Dynamics Istra4D software. This program allows us to analyze interesting values of the whole element or its fragment visible on the camera. An example image of the stent model, with a special speckle pattern, from the camera as seen in Istra4D is shown in Figure 15.

The compression tests were performed as quasi-static with the force-displacement (crosshead displacement) diagram presented in Figure 16.

The Istra4D program allows us to apply a mask to the area of interest and then generate deformation maps on it, among other things. In this way, experimental results were obtained for both longitudinal and transverse compression. Figure 17 presents images from the Istra4D program showing the compressed stent models with a mask applied to the sections whose deformation results at the assumed force obtained by the experimental method we will compare with the results of the numerical simulation at the same boundary conditions.

In the next step, Istra4D allows us to obtain the results we are interested in from a series of images taken with the camera and imported into the software. It is possible to obtain the results of stress, strain, or as in this case, displacements in the area marked with a mask. The results are presented in the form of color maps.

## 5. Results Obtained by Numerical Simulations and Experimental Studies

We obtained the results of the analysis of the stent fragment selected as representative of the stent model by both numerical simulations and experimental studies. Figure 18 shows a comparison of the results for transverse stent compression (pinch) for the numerical simulation as well as the experimental study developed using DIC Istra4D. The unit of displacements is mm.

The results allow one to compare displacements as the results of transverse compression (pinch) of the stent obtained from numerical simulation using the FEM package MSC Marc and measured using the DIC system. Looking at the stent sidewall section that was speckled for DIC and then masked in DIC software Istra4D, we can see that the displacements are in the range between 0.62 and 0.76 mm. Since an identical force was applied in the numerical simulation, observing the same stent section that was subjected to the experimental study, we can observe that the displacements range from 0.65 to 0.81 mm.

Figure 19 shows a comparison of the results for longitudinal stent compression for the numerical simulation and the experimental study.

The results of compressing the stent with a longitudinal force showed good agreement between numerical simulation and experimental measurements. Observing the same section as in transverse compression, we see that the results from DIC are between 0.22 mm and 0.34 mm, and under the same boundary conditions, the displacements of the stent of the same fragment of the analyzed using FEM in the MSC Marc software are in the range 0.27 mm–0.32 mm.

The real stent produced using injection moulding does not have the same shape as the numerical model; moreover, during the process the material density was not perfectly uniform in the stent, leading to differences between the measured and simulated results. The difference between measured values below 7% shows that numerical simulations can be used with great success for the stent design and performance measurement for both tests. The numerical simulations performed indicate that the stent retains adequate stiffness under operating conditions, taking into account possible loading and deformation states that result primarily from the implantation process. The obtained results are also confirmed by the experimental tests conducted. The application of the DIC method significantly expands the research possibilities, allowing for a detailed inspection of the deformation state and, above all, verification of local dangerous areas (rounded parts of the crown) in terms of loss of continuity of the structure, especially in the crimped state. This approach significantly increases the possibility of assessing the quality of the stent.

## 6. Conclusions

In the first part of this article, an original method for stent design was presented using two-stage modeling. It was a convenient way to visualize the original appearance and selection of the geometrical parameters of the stents, which are produced in expanded form. In addition, this approach helps to choose the right geometry of the expanded stent, which is the most important step in designing the crimping process. The numerical simulation of the reverse process to the crimping process is subject to a smaller error during implementation. This is because the outer surface of the catheter, which is expanding, is still in contact with the inner wall of the stent. Even if it is deformed at some point, we still have contact points that allow continuous operation of the process. This is not the case for the crimping process. In the case of excessive deformation of the FEM mesh of the stent model, the inner surface of the rigid plate cannot make contact with the stent and the simulation is interrupted. This means, of course, that the geometry is not properly selected, but it does not give us information about what should be corrected to obtain the right geometry. Put simply: even inappropriate geometries can be opened and drawn on the basis of simulations the conclusions on the optimization of the expanded form of the stent, but the inadequate geometry cannot be closed. Therefore, the reverse approach gives valuable information about the type of stent we are interested in: open, as it is produced by different techniques.

Furthermore, for the selected stent, we obtained information about the deformations and stress that occur during crimping and deployment. For both cases, the stresses are within an acceptable range. As can be seen from the deformation state of the stent recognized as optimal, there is no crown bending, as it has in the case of a longer ring. Such a deformation of the stent can lead to the stent being inadequate in its function: it will not support the diseased artery in which it would be placed, it would have too small a radial force. Furthermore, such a deformation of the geometry of the open stent could lead to a puncture of the blood-vessel wall. A stent with such a geometry could also break in the patient’s body, and looking at the decomposition time of 12 to 24 months could cause great damage before it undergoes a biodegradation process. There is also no crown extension, making the stent impossible to perform. Furthermore, a stent with such a ring length could break during opening, making it unable to support a diseased artery, and in addition, its fragments would be in the patient’s body for several months before they would degrade.

Regarding the second part of the paper, in which we compare the results of the numerical simulation with experimental studies of the fabricated stent model, we can observe that the displacements of the selected fragment are comparable. This is the case for both transverse compression of the stent (pinch) and longitudinal compression of the stent.

The results of numerical simulations and experimental tests are promising; the discrepancies between the results are small, with the error of displacement below 7%.

## Figures and Tables

**Figure 1 materials-15-02385-f001:**
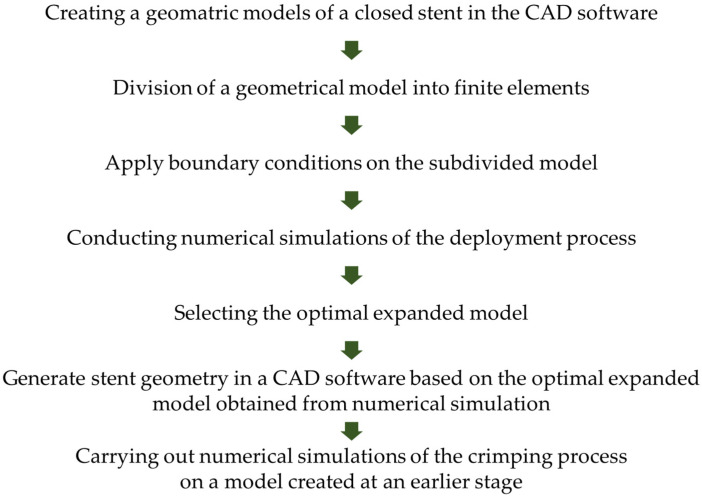
Diagram of two-step modeling in a given case.

**Figure 2 materials-15-02385-f002:**
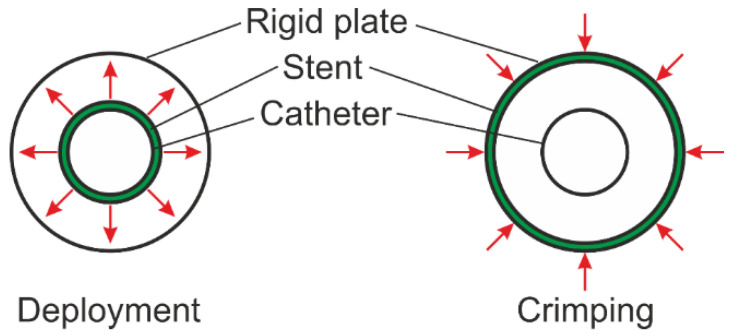
Diagram presenting the procedure during numerical simulations.

**Figure 3 materials-15-02385-f003:**
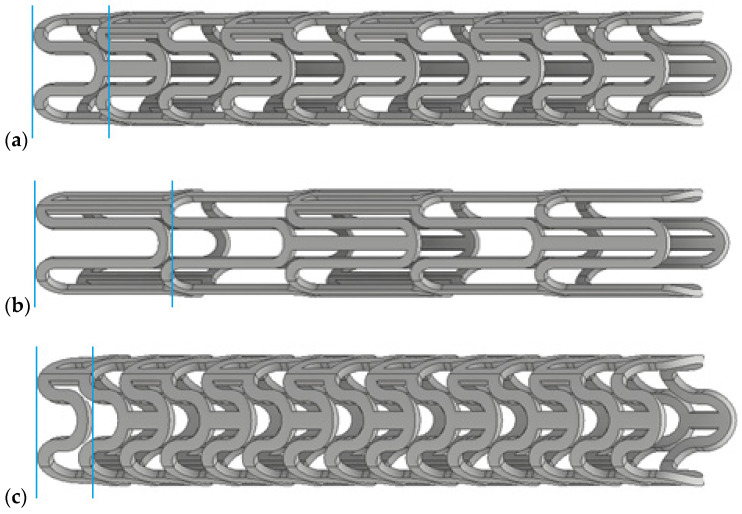
Geometric models of stents, for which crown length is marked between the two sections and is: (**a**) optimal, (**b**) too wide, and (**c**) too narrow.

**Figure 4 materials-15-02385-f004:**
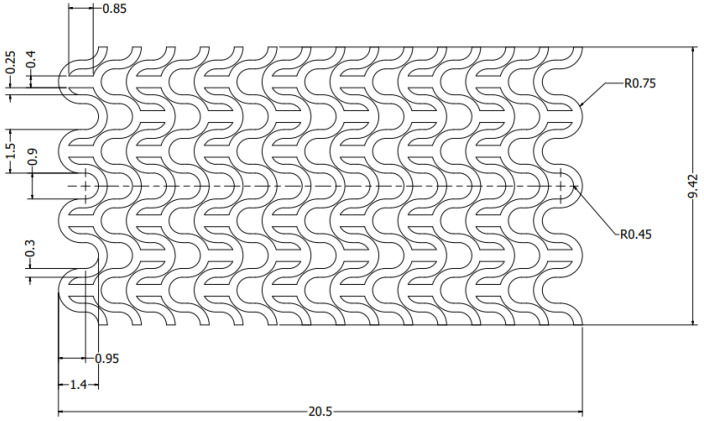
Sketch of the half of the geometry of the stent.

**Figure 5 materials-15-02385-f005:**
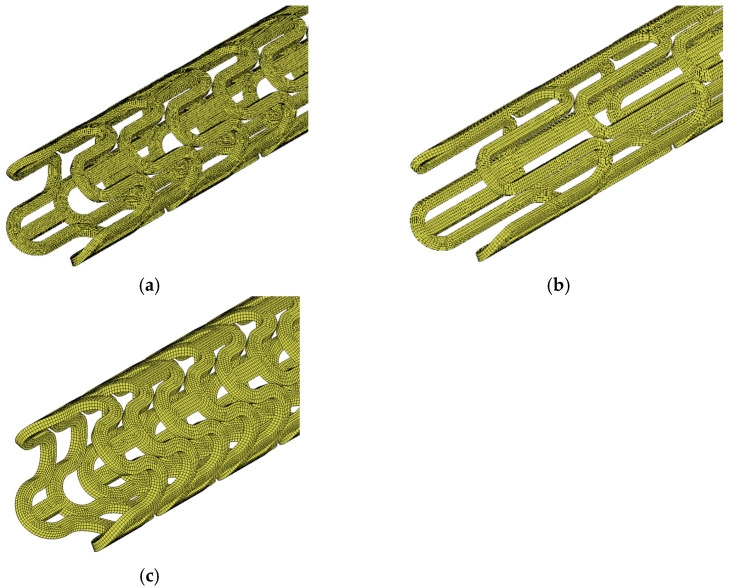
Discretized models of stents: (**a**) optimal, (**b**) too long, (**c**) too short.

**Figure 6 materials-15-02385-f006:**
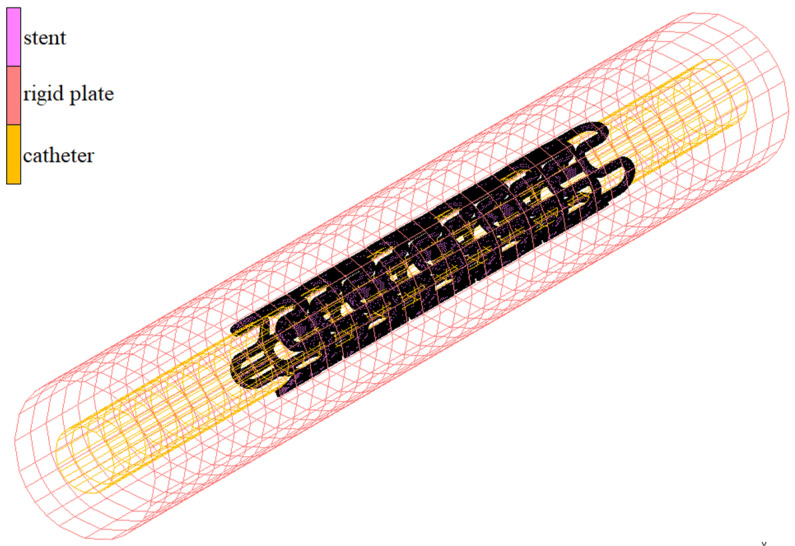
Numerical model for deployment simulation.

**Figure 7 materials-15-02385-f007:**
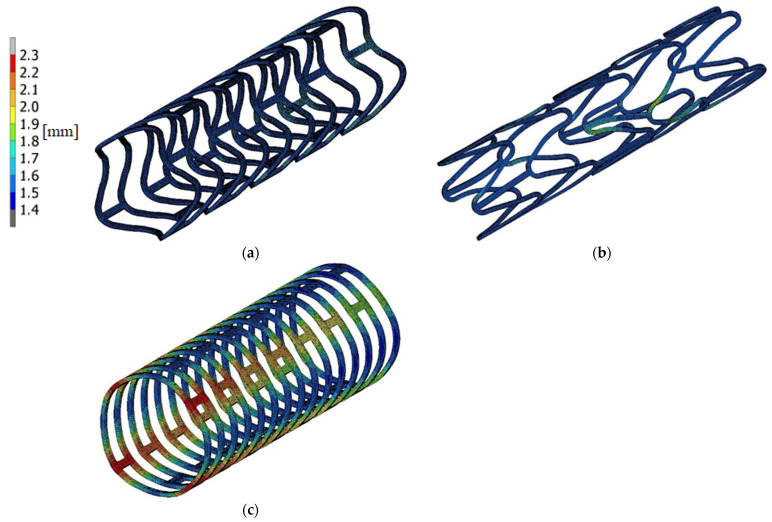
The deformation state after deployment for stents: (**a**) optimal, (**b**) too long, (**c**) too short.

**Figure 8 materials-15-02385-f008:**
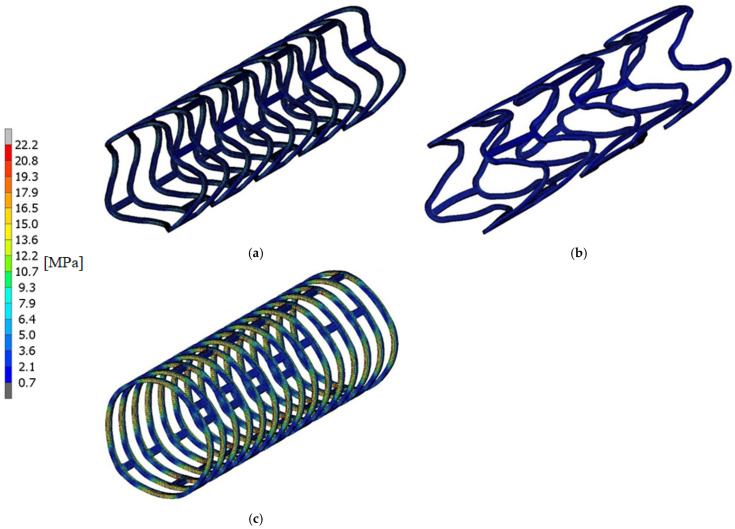
Stress state after stent deployment: (**a**) optimal, (**b**) too long, (**c**) too short (von Mises).

**Figure 9 materials-15-02385-f009:**
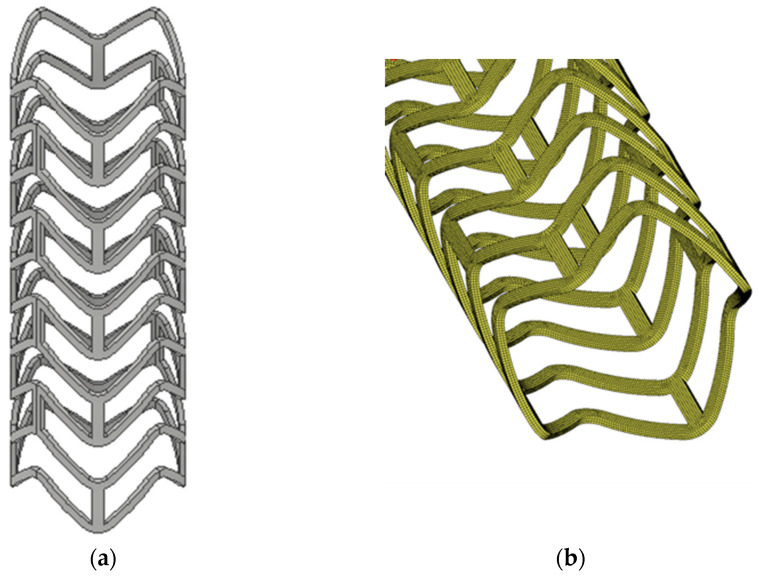
The modified stent model after deployment simulation selected for the crimping process: (**a**) geometry, (**b**) discretized using finite elements.

**Figure 10 materials-15-02385-f010:**
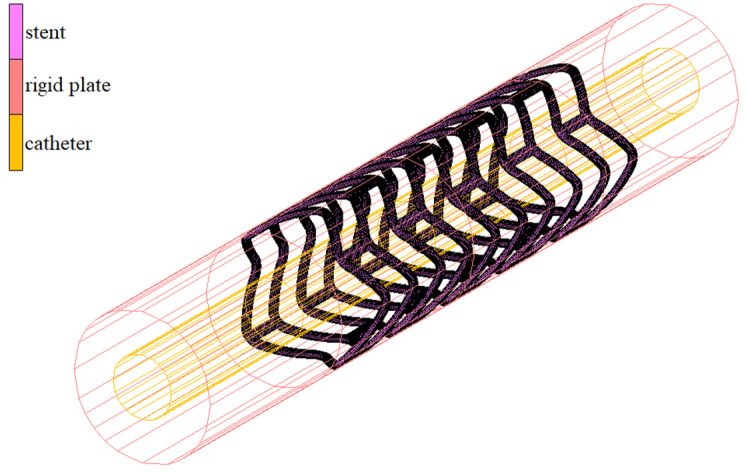
Numerical model for crimping simulation.

**Figure 11 materials-15-02385-f011:**
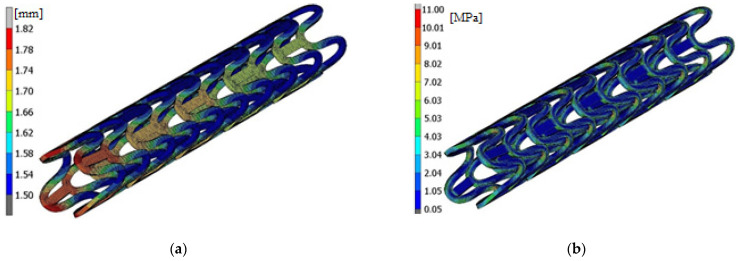
State after the stent crimping: (**a**) the deformation, (**b**) the stress (von Mises).

**Figure 12 materials-15-02385-f012:**
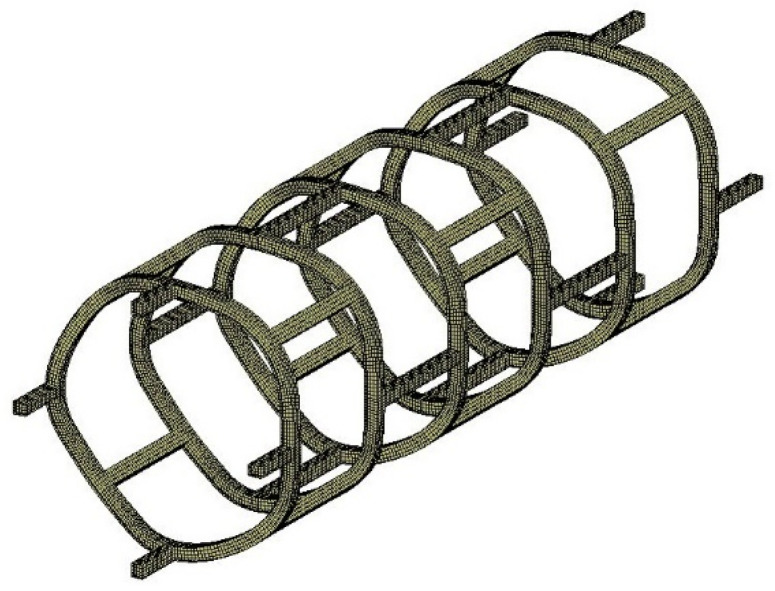
Prototype geometry after discretization.

**Figure 13 materials-15-02385-f013:**
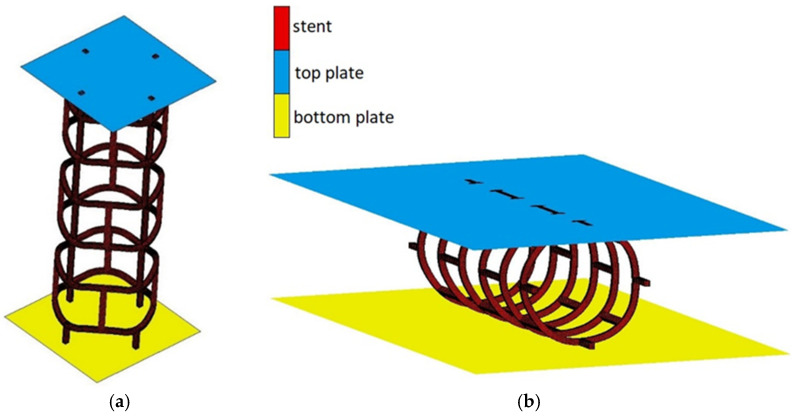
Stent compression: (**a**) longitudinal, (**b**) transverse.

**Figure 14 materials-15-02385-f014:**
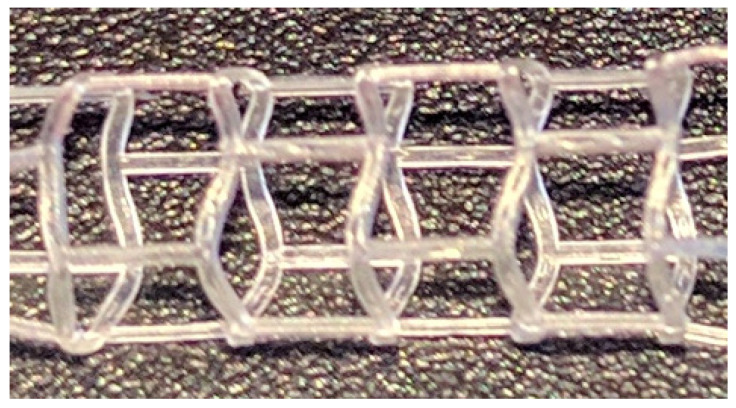
Model fabricated by microinjection moulding.

**Figure 15 materials-15-02385-f015:**
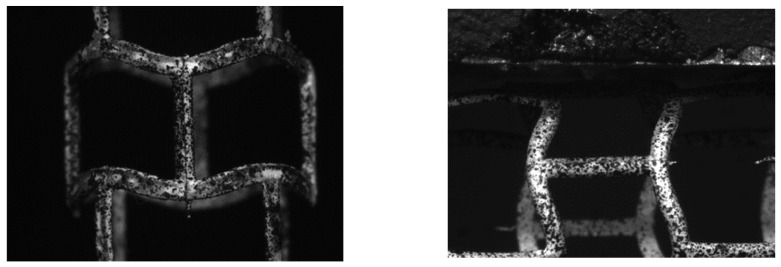
Stent model covered with speckle pattern.

**Figure 16 materials-15-02385-f016:**
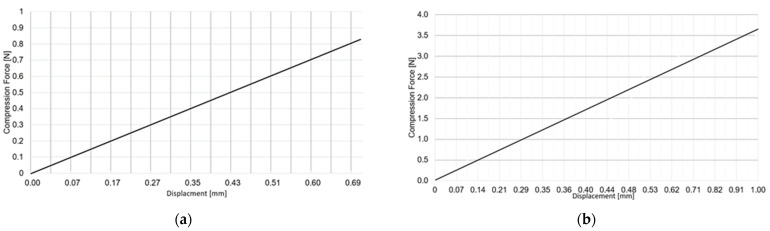
Force-displacement diagram for: (**a**) transverse and (**b**) longitudinal compression test.

**Figure 17 materials-15-02385-f017:**
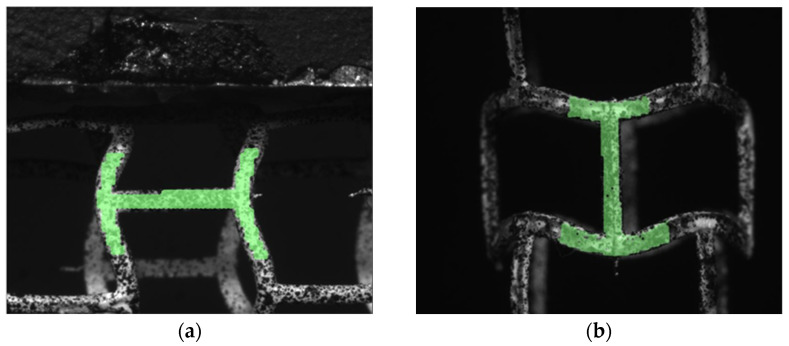
Istra4D DIC masked stents: (**a**) transversely compressed (pinch), (**b**) longitudinally compressed.

**Figure 18 materials-15-02385-f018:**
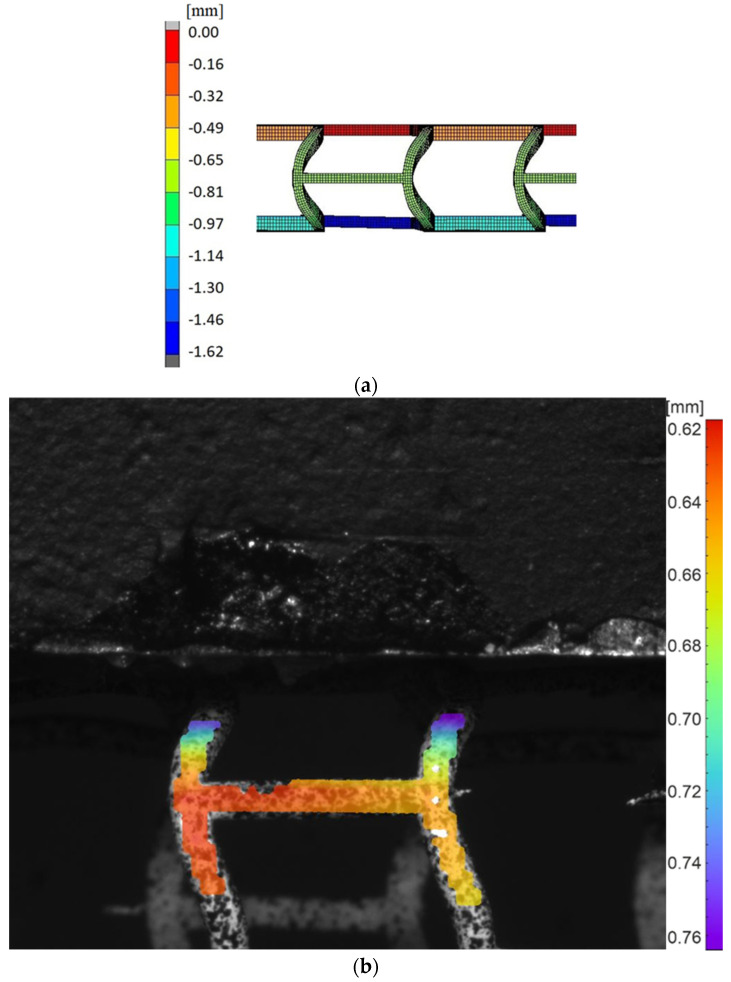
Displacement maps for transverse compression (pinch): (**a**) simulated using FEM, (**b**) measured using DIC.

**Figure 19 materials-15-02385-f019:**
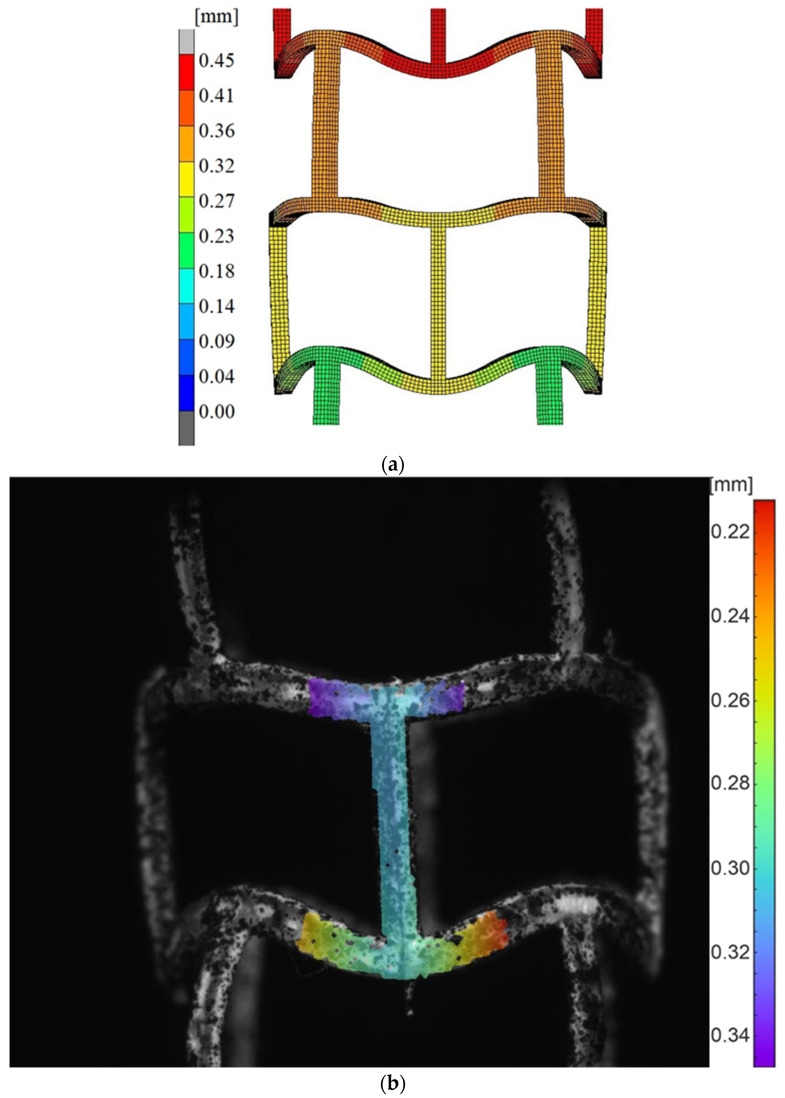
Displacement maps for longitudinal compression: (**a**) simulated using FEM, (**b**) measured using DIC.

## Data Availability

The data presented in this study are available on request from the corresponding author.

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
