# Peer review of "Two-Step Geometry Design Method, Numerical Simulations and Experimental Studies of Bioresorbable Stents"

_materials, 2022, doi:10.3390/ma15072385_

Round 1

Reviewer 1 Report

It is a great honor to review this manuscript. The author's work is interesting, but I would like the authors to respond to the comments below.

1.The abstract section should give more data.

2.Why both the section 2 and section 3 make introduction about materials?

3.How do you get the three initial geometries of contracted stents? There are a variety of stents with different shapes and textures now, why did you choose this one? (line 173)

4.Why these values remained the same in all cases? (line 181) They have an important influence on the mechanical properties of the stents.

5.The title of Figure 3 needs to clearly indicate what is long and what is short. (line 189)

6.How did you determine the initial parameters? For example, the thickness of the stent was 0.2 mm. (line 191)

7.In the process of setting boundary conditions, why did the authors not consider the force of blood flow on the inner surface of the stent? (line 209)

8.Units such as MPa, mm are missing on the simulation result graphs.

9.The transverse compression force setting is unreasonable. The stent should be subjected to compressive force for 360°, while Fig. 13(b) is only forced up and down.

10.Can the authors give a force-displacement curve during the experimental test?

11.The stents are round. However, the results in Fig. 16(a) represent the deformation in the vertical direction. As shown in the below figure.

(Please see the PDF attachment)

12.In the section 5, the authors only describe the experimental phenomenon, and cannot give a deep explanation.

13.Do the authors need to consider the effects of time and dissolution rate in the design process? Bioresorbable stents will gradually dissolve in the blood, and during this process, the mechanical properties of the stent will decrease.

14.Why did the authors only consider compression performance in the design of the stent? Are there no other important parameters to consider?

15.In the process of the author's design, the type of the stent is single. Can this design method be applied to other types of stent designs? For example, a stent formed by winding, or a stent that is inconsistent with the engraved shape in this manuscript?

Reviewer 2 Report

I have reviewed the manuscript by MolÄ™da et al., “Two-Step Geometry Design Method, Numerical Simulations and Experimental Studies of Bioresorbable Stents” submitted to “Materials” for publication. In this study, the authors have proposed an original two-step procedure for designing the stent geometry and investigated numerical simulations and deployment processes. The manuscript fits well with in the scope of the journal, however needs some further improvements; there are a few suggestions that authors may consider to improve it further:

The use of English language is reasonable, however, there are a number of punctuation and grammatical errors; that should be corrected and rephrased using academic English for a better flow of text for reader.

Abstract: is reasonable, however some of the key findings/results can be further added to improve the abstract.

Authors should make sure that all the abbreviations are defined at their first appearance in the text and use abbreviations afterword.

Similarly, please subscript numeric in the chemical formula; for example H2O: 2 should be subscripted.

Introduction is very comprehensive and detailing all the background information and rationale of the study. In my opinion, it is too long and detailed for an orgiical paper, therefore, some of the less relevant information can be deleted.

Figures 7 and 8 should be merged as one figure for the comparison purpose.

Similarly, authors may consider merging some other figures too in order to reduce total number of figures

Results and discussion sections are comprehensive, coving all the results and their discussion.

A separate section of conclusions briefing key finding can be added.

Round 2

Reviewer 1 Report

It seems the authors have addressed the related issues. There are still some minor issues to be aware of. Such as in Figure 16, There should be no commas between numbers. (0,1-0.1, 0,07-0.07)

Author Response

Thank you very much for your observation; the decimal separator in Figure 16 has been corrected. This was included in the paper.

We would also like to thank you for your comments during the review process, which allowed us to improve the article.

Reviewer 2 Report

The authors' revision and response to the comments is satisfactory.

thank you. 

Author Response

We would also like to thank you for your comments during the review process, which allowed us to improve the article.